# Risk Prediction of Second Primary Endometrial Cancer in Obese Women: A Hospital-Based Cancer Registry Study

**DOI:** 10.3390/ijerph18178997

**Published:** 2021-08-26

**Authors:** Chi-Chang Chang, Chun-Chia Chen, Chalong Cheewakriangkrai, Ying Chen Chen, Shun-Fa Yang

**Affiliations:** 1School of Medical Informatics, Chung Shan Medical University and IT Office, Chung Shan Medical University Hospital, Taichung 40201, Taiwan; changintw@gmail.com (C.-C.C.); amy0988147957@gmail.com (Y.C.C.); 2Department of Information Management, Ming Chuan University, Taoyuan 33300, Taiwan; 3Institute of Medicine, Chung Shan Medical University, Taichung 40201, Taiwan; ysf@csmu.edu.tw; 4Department of Surgery, Division of Plastic Surgery, Chung Shan Medical University Hospital, Taichung 40201, Taiwan; 5Department of Obstetrics and Gynecology, Division of Gynecologic Oncology, Faculty of Medicine, Chiang Mai University, Chiang Mai 50200, Thailand

**Keywords:** second primary cancers (SPCs), endometrial cancer (EC), second primary endometrial cancer (SPEC), risk prediction

## Abstract

Due to the high effectiveness of cancer screening and therapies, the diagnosis of second primary cancers (SPCs) has increased in women with endometrial cancer (EC). However, previous studies providing adequate evidence to support screening for SPCs in endometrial cancer are lacking. This study aimed to develop effective risk prediction models of second primary endometrial cancer (SPEC) in women with obesity (body mass index (BMI) > 25) and included datasets on the incidence of SPEC and the other risks of SPEC in 4480 primary cancer survivors from a hospital-based cancer registry database. We found that obesity plays a key role in SPEC. We used 10 independent variables as predicting variables, which correlated to obesity, and so should be monitored for the early detection of SPEC in endometrial cancer. Our proposed scheme is promising for SPEC prediction and demonstrates the important influence of obesity and clinical data representation in all cases following primary treatments. Our results suggest that obesity is still a crucial risk factor for SPEC in endometrial cancer.

## 1. Introduction

Endometrial cancer (EC) is the most common gynecological malignancy, and its incidence is rising alongside the growing prevalence of obesity [1]. Endometrial cancer affects women worldwide, resulting in an estimated 42,000 deaths annually [2]. EC most commonly occurs after menopause, related to long-term exposure to unopposed estrogens. On average, the overall 5-year survival rate is around 80%. Overweight (defined as body mass index (BMI) of at least 25 kg/m^2^) also represents an important risk factor in 50% of endometrial cancers. A BMI above 25 kg/m^2^ doubles a woman’s risk of endometrial cancer, and a BMI above 30 kg/m^2^ triples the risk [3,4]. Therefore, understanding the key mechanisms driving endometrial carcinogenesis in primary endometrial cancer (PEC) may affect second primary endometrial cancer (SPEC) diagnoses if aimed at those at highest risk. An understanding of the correlation between obesity and SPEC is critical in developing such prevention strategies [1].

In the Taiwan Cancer Registry database, nine variables are recorded as clinical prognostic factors of EC: (1) age at diagnosis, (2) grade/differentiation, (3) tumor size, (4) clinical stage group, (5) pathologic stage group, (6) surgical margin involvement at the primary site, (7) date of first surgical procedure, (8) sequence of radiotherapy and surgery, and (9) sequence of locoregional therapy and systemic therapy. In this study, we hypothesized that these factors and BMI are important predictors of SPEC in endometrial cancers. Therefore, the purpose of the analysis was to identify the most important risk factors from the 10 predictors listed in Table 1 and Table 2.

We suggest potential prevention strategies and demonstrate the need for risk prediction models that identify specific groups of women at particularly high risk of endometrial cancer, for whom risk-reducing interventions are likely to have a significant impact.

## 2. Materials and Methods

A hospital-based cohort of 4480 patients diagnosed with endometrial cancer was identified from the database of the Taiwan Cancer Registry from 2009 to 2016. The risk of endometrial cancer in age- and grade-deferential, clinical or pathological stages or therapies was compared using analysis of obese and non-obese groups. Using these different decision tree models, prediction factor combinations for conditions of interest were identified. Moreover, a comprehensive clinical prevention approach was associated with all factors.

We aimed to use data mining methods including support vector machine (SVM), linear discriminant analysis (LDA), logistic regression (LGR), C4.5, classification and regression tree (CART), random forest (RF), and C5.0 to predict second primary endometrial cancer in obese women with different variables (Table 3). The classification accuracy of the seven methods was evaluated using receiver operating characteristic curve analysis to estimate the area under the curve (AUC) (Table 4). Accuracy, sensitivity, and specificity were considered in this study (Figure 1). SVM classifiers operate by separating two classes using a linear decision boundary called the hyperplane. The hyperplane places data to maximize the distance between the hyperplane and instances [5,6]. LDA is a supervised leaning algorithm used for dimensionality reduction and classification. It also uses a feature extraction and data compression technology [7,8]. LGR is the most widely used modeling approach for binary outcomes in epidemiology and medicine. The model is part of the family of generalized linear models that explicitly model the relationship between explanatory variable X and response variable Y [9,10]. The C4.5 decision tree is a common and excellent machine algorithm that selects the decision tree’s attributes on each node based on the concept of information entropy. It adopts a greedy approach in which the decision trees are constructed in a top-down recursive divide-and-conquer manner [11,12]. RF is an ensemble learning method. It generates many classification trees by selecting subsets of the given dataset and selecting subsets of predictor variables randomly, finally aggregating the results of all models to obtain a random forest [13]. The C5.0 decision tree is a classification approach that generates the tree in a top-down scheme based on the given information using a recursive process [14]. CART is a decision tree system that uses a binary recursive procedure to partition the data in homogenous subsets based on the Gini index. The CART algorithm classifies data in the process. The classification process is similar to a tree structure, including root, node and leaf. [15,16].

Several researchers have studied the use of machine learning technologies in developing predictive models for cancers. Shih et al. [17] utilized LDA, C4.5 decision trees, and CART to predict early chronic kidney disease in patients. Tseng et al. [18] investigated the use of SVM to predict the recurrence of cervical cancer. Tseng et al. [19] reported on the use of SVM and RF in predicting risk factors and the recurrence of ovarian cancer. The important variables and coding data in Table 3, which were collected by the Taiwan hospital registry database, were used in this study. Based on the literature and discussion with clinicians, we used 10 independent variables that were determined as the risk factors for SPEC as predicting variables.

With the highest AUC value, CART produced an ideal prediction model for the obese women (BMI > 25) (Figure 2) in this study.

## 3. Results

During the study period, 520 patients were diagnosed with SPEC in primary endometrial cancers. Figure 2 shows the CART classification tree depicting the SPCs of endometrial cancer predictors. For CART decision tree stratification, the status of the branches of the tree is based on the priority of all independent variables.

All subjects were divided into 11 subgroups, from the root node to leaf nodes, through different branches. As previously explained, the pathologic stage variable has a strong influence on the interpretation of the SPEC and was therefore identified as the root node of the classified decision tree.

The first-rule decision tree was obtained from the following determining factors: pathologic stage (<II) and surgical margin involvement (Yes); the accuracy obtained was 1.0 across 17 samples. The second-rule decision tree was obtained from the following determining factors: pathologic stage (<II), surgical margin involvement (No), tumor size (<2 cm), clinical stage (≥II), and age at diagnosis (≥50); the accuracy obtained was 1.0 across 32 samples. The fourth-rule decision tree was obtained from the following determining factors: pathologic stage (<II), surgical margin involvement (No), tumor size (<2 cm), clinical stage (<II), and sequence of radiotherapy/surgery (Yes); the accuracy obtained was 0.882 across 17 samples. The fifth-rule decision tree was obtained from the following determining factors: pathologic stage (<II), surgical margin involvement (No), tumor size (<2 cm), clinical stage (<II), sequence of radiotherapy/surgery (No), and sequence of locoregional/systemic therapy (Yes); the accuracy obtained was 0.7 across 20 samples. The eighth-rule decision tree was obtained from the following determining factors: pathologic stage (<II), surgical margin involvement (No), tumor size (≥2 cm), age at diagnosis (<50), sequence of locoregional/systemic therapy (Yes), and clinical stage (≥II); the accuracy obtained was 1.0 across 16 samples. Therefore, the decision tree could be divided into abnormal (ABNL; SPEC) or normal (NL; non-SPEC) situations. The accuracy ranged from 68.5% to 100% (Figure 2). Five rules are related to the prediction models of SPEC in endometrial cancer in obese women (Table 5).

For obese women (BMI > 25 kg/m^2^), age (≥50 years, *p* = 0.019), tumor size (≥2 cm, *p* < 0.001), clinical stage and pathological stage (<II, *p* < 0.001), surgery (Yes, *p* = 0.014), and sequence of radiotherapy/surgery (No, *p* < 0.001) increased the risk of SPEC in endometrial cancer (Table 6).

## 4. Discussion

Recent advances in diagnostic and therapeutic methods have increased the overall survival rate of patients with cancers. As cancer survival rates have increased, the incidence of second primary cancers has gradually increased. However, this phenomenon is due to multiple factors such as genetic or environmental factors and the development of new anti-cancer drugs. In the present study, SPEC in endometrial cancers was observed in 11.6% of 4480 patients who had ever been diagnosed with primary endometrial cancer. Obesity (BMI ≥ 30 kg/m^2^) is the strongest risk factor for primary EC. For every 5 kg/m^2^ increase in BMI, there is a 60% increased risk of EC, with a BMI above 25 kg/m^2^ doubling the risk and a BMI above 30 kg/m^2^ tripling the risk [20]. However, obesity may not be a crucial risk factor in second primary endometrial cancers [21].

Currently, there is no benefit to early screening for endometrial cancer as screening is unable to decrease mortality from endometrial cancers; it mainly detects women with low-risk tumors [22]. In literature reviews, increasing age and long-term exposure to unopposed estrogens are strong risk factors for endometrial cancer. Metabolic syndrome (obesity, diabetes) is also a well-known risk factor. It alters the concentrations of insulin-like growth factor and its binding proteins [23]. Estrogen receptor transcriptional activity can be induced by signaling by insulin-like growth factor 1 even in the absence of estradiol, which increases the incidence of endometrial cancer [24,25,26,27]. In our study, obesity seems to be an independent risk factor of primary endometrial cancer. It also plays a key role in the incidence of second primary endometrial cancer [28].

The use of preoperative radiotherapy has been abandoned because it interferes with surgical staging and there is no benefit compared to postoperative radiotherapy [1]. The aim of adjuvant radiotherapy is the pelvic lymph-node regions that might contain microscopic metastasis, as well as the central pelvic region and the upper vagina. There is a consensus that patients with lesions of surgical stage IA or IB and grade 1 or 2 (low risk) can be treated without postoperative radiotherapy [29]. Isolated pelvic and vaginal recurrences of low-risk endometrial cancers can be successfully treated at the time of recurrence without radiotherapy. Therefore, radiotherapy is usually used in advanced endometrial cancer. In our study, postoperative radiotherapy was found to be an increasing risk factor in the non-obesity group but a decreasing risk factor in the obesity group.

Endometrial cancer is a surgically staged disease. The most important therapy for endometrial cancer is surgery. Surgical staging provides prognostic information for survivors. In our study, most patients (99.31%) had received surgical intervention for their endometrial cancer. All second primary endometrial cancer was from these patients. In our study, for the obesity group, one early endometrial cancer (stage < II) case who had received surgery without radiotherapy and systemic therapy had a higher risk of second primary endometrial cancer at older age (≥50 years).

In the past, we successfully used data mining classification techniques for building a predictive model of early chronic kidney disease [17]. In this study, we successfully applied 10 prognostic factors to determine SPEC risk factors in obese women using data mining algorithms. However, there might be some limitations from using only Taiwan’s local hospital registry database, which may not represent other ethnicities. The tree-based algorithm was dependent on local consensus to decide the variables in the predictive modeling. International database pooling analysis was suggested for future studies. Some clinic-pathological factors such as histological type, family history of cancer, timing of chemotherapy exposure, and regimen used should be included in future analyses. The strength of our current study was a comprehensive Taiwan hospital registry database. Our promising results could guide us to create another possible predictive model for other gynecologic cancers in the future.

## 5. Conclusions

Age (>50 years), BMI (>25 kg/m^2^), grade/differentiation, cancer stage, grade, and adjuvant therapies were used as prognostic factors of endometrial cancer. In our study, we found these factors can be used to predict second primary endometrial cancer. Obesity is an independent risk factor of second primary endometrial cancer.

Obese women have a higher risk of endometrial cancer. In this study, the decision tree could be divided into abnormal (SPEC) or normal (non-SPEC) situations in obese women with primary endometrial cancer, with accuracy ranging from 68.5% to 100%. In obese women, we also identified that age at diagnosis, tumor size, clinical stage and pathological stage, surgery, and the sequence of radiotherapy had important impacts on the predictivity of the models, whereas other predictors, such as grade/differentiation, surgical margin involvement and locoregional/systemic therapy, were less important.

## Figures and Tables

**Figure 1 ijerph-18-08997-f001:**
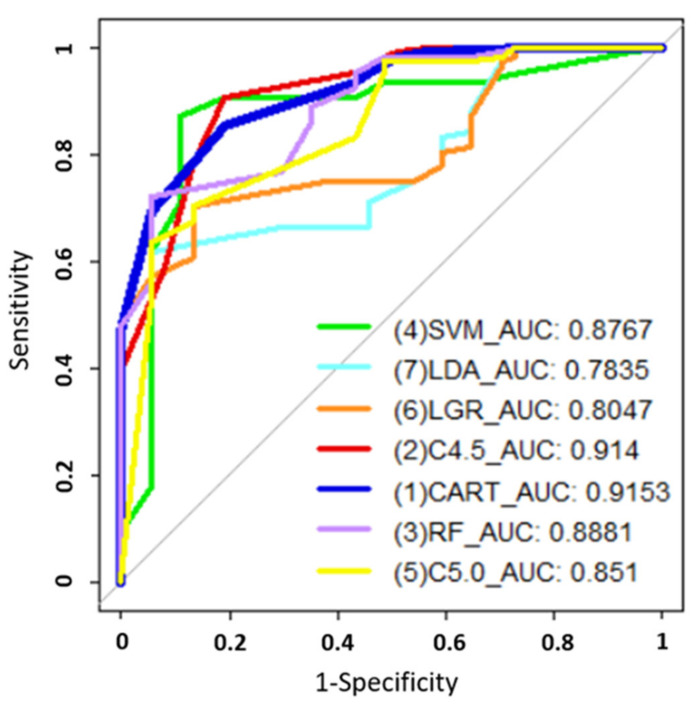
Receiver operating characteristic (ROC) curves of the seven methods with AUCs.

**Figure 2 ijerph-18-08997-f002:**
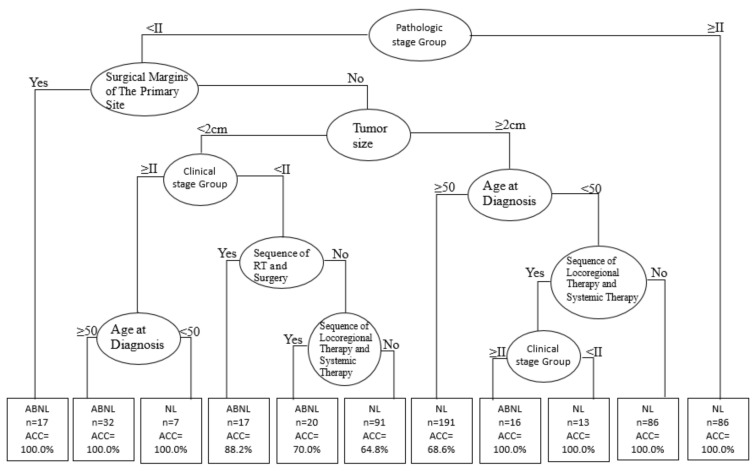
CART classification tree depicting the SPEC of primary endometrial cancer predictors (BMI > 25). ABNL: abnormal/diagnosed with SPEC; NL: Normal/diagnosed with non-SPEC; ACC: Accuracy.

**Table 1 ijerph-18-08997-t001:** The important variables associated with endometrial cancer.

Rank	Variable Name
1	Clinical stage group
2	Tumor size
2	Pathologic stage group
2	Date of first surgical procedure
5	BMI
6	Age at diagnosis
7	Sequence of locoregional therapy and systemic therapy
8	Grade/differentiation
8	Surgical margins of the primary site
10	Sequence of RT and surgery

**Table 2 ijerph-18-08997-t002:** Subject demographics of all primary endometrial cancers.

Characteristics	Endometrial Cancer (N = 1560)		
	Without SPEC	With SPEC	*p*-Value
N (%)	1040 (66.7%)	520 (33.3%)	
Age at Diagnosis (years)			<0.001 **
<50	372 (35.7%)	140 (26.9%)	
≥50	668 (64.3%)	380 (73.1%)	
Grade/Differentiation			0.014 *
1, 2	705 (67.8%)	320 (61.5%)	
Others	335 (32.2%)	200 (38.5%)	
Tumor Size (cm)			<0.001 **
<2	262 (25.2%)	220 (42.3%)	
≥2 c	778 (74.8%)	300 (57.7%)	
Clinical Stage			<0.001 **
<II	838 (80.6%)	280 (53.8%)	
≥II	202 (19.4%)	240 (46.2%)	
Pathologic Stage			<0.001 **
<II	834 (80.2%)	480 (92.3%)	
≥II	206 (19.8%)	40 (7.7%)	
Surgical Margin Involvement			0.405
No	947 (91.1%)	480 (92.3%)	
Yes	93 (8.9%)	40 (7.7%)	
Surgical Procedure			<0.001 **
No	40 (3.8%)	0 (0.0%)	
Yes	1000 (96.2%)	520 (100.0%)	
Sequence of Radiotherapy/Surgery			0.689
No	611 (58.8%)	300 (42.9%)	
Yes	429 (41.2%)	220 (57.1%)	
Sequence of Locoregional/Systemic Therapy			<0.001 **
No	740 (71.2%)	320 (57.7%)	
Yes	300 (28.8%)	200 (42.3%)	
BMI (kg/m^2^)			0.001 *
≤25	464 (44.6%)	280 (53.8%)	
>25	576 (55.4%)	240 (46.2%)	

** *p*-value < 0.001; * *p*-value < 0.05, calculated by simple chi-squared tests.

**Table 3 ijerph-18-08997-t003:** Important variables and coding in this study.

Variable	Name	Definition of Normal Test Data
X1	Age at diagnosis (years)	<50/≥50
X2	Grade/differentiation	≤2/>2
X3	Tumor size (cm)	<2/≥2
X4	Clinical stage group	<II/≥II
X5	Pathologic stage group	<II/≥II
X6	Surgical margins of the primary site	No/Yes
X7	Date of first surgical procedure	No/Yes
X8	Sequence of RT and surgery	No/Yes
X9	Sequence of locoregional therapy and systemic therapy (chemotherapy)	No/Yes
X10	BMI (kg/m^2^)	≤25/>25
Y	SPEC	No/Yes

**Table 4 ijerph-18-08997-t004:** Classification results of the seven methods with area under the curve (AUC).

Method	Accuracy	Sensitivity	Specificity	AUC
SVM	0.875	0.8919	0.8692	0.8767
LDA	0.7014	0.9459	0.6168	0.7835
LGR	0.7431	0.8649	0.7009	0.8047
C4.5	0.8819	0.9065	0.8108	0.914
CART	0.8403	0.8108	0.8505	0.9153
RF	0.7778	0.9459	0.7196	0.8881
C5.0	0.7153	0.9459	0.6355	0.851

**Table 5 ijerph-18-08997-t005:** The summarized rules of condition variables (BMI > 25 kg/m^2^).

Rules No.	Combinations of Condition Variables	SPEC/Observed (n)	Accuracy
**1**	Pathologic stage (<II) + Surgical Margins involvement (Yes)	17/20	100.0%
**2**	Pathologic stage (<II) + Surgical Margins involvement (No) + Tumor size (<2 cm) + Clinical stage (≥II) + Age at Diagnosis (≥50)	32/40	100.0%
**4**	Pathologic stage (<II) + Surgical Margins involvement (No) + Tumor size (<2 cm) + Clinical stage (<II)+ Sequence of Radiotherapy (Yes)	17/24	88.2%
**5**	Pathologic stage (<II) + Surgical Margins involvement (No) + Tumor size (<2 cm) + Clinical stage (<II) + Sequence of Radiotherapy (No) + Sequence of Locoregional/Systemic Therapy (Yes)	20/28	70.0%
**8**	Pathologic stage (<II) + Surgical Margins involvement (No) + Tumor size (≥2 cm) + Age at Diagnosis (<50) + Sequence of Locoregional/Systemic Therapy (Yes) + Clinical stage (≥II)	16/20	100.0%

**Table 6 ijerph-18-08997-t006:** The subject demographics of independent predictors of SPEC in primary endometrial cancer.

Characteristics	BMI ≤ 25 kg/m^2^			BMI > 25 kg/m^2^		
	Without SPEC	With SPEC	*p*-Value	Without SPEC	With SPEC	*p*-Value
N (%)	560 (66.7%)	280 (33.3%)		480 (66.7%)	240 (33.3%)	
Age at Diagnosis			0.117			0.019 *
<50 years	190 (33.9%)	200 (71.4%)		161 (33.5%)	60 (25.0%)	
≥50 years	370 (66.1%)	80 (28.6%)		319 (66.5%)	180 (75.0%)	
Grade/Differentiation			0.004			0.698
1, 2	377 (67.3%)	160 (57.1%)		313 (65.2%)	160 (66.7%)	
Others	183 (32.7%)	120 (42.9%)		167 (34.8%)	80 (33.3%)	
Tumor Size			0.144			<0.001 **
<2 cm	172 (30.7%)	100 (35.7%)		102 (21.3%)	120 (50.0%)	
≥2 cm	388 (69.3%)	180 (64.3%)		378 (78.7%)	120 (50.0%)	
Clinical Stage			<0.001			<0.001**
<II	463 (82.7%)	120 (42.9%)		385 (80.2%)	160 (66.7%)	
≥II	97 (17.3%)	160 (57.1%)		95 (19.8%)	80 (33.3%)	
Pathologic Stage			0.032			<0.001 **
<II	446 (79.6%)	240 (85.7%)		384 (80.0%)	240 (100.0%)	
≥II	114 (20.4%)	40 (14.3%)		96 (20.0%)	0 (0.0%)	
Surgical Margins Involved			0.648			0.170
No	515 (92.0%)	260 (92.9%)		424 (88.3%)	220 (91.7%)	
Yes	45 (8.0%)	20 (7.1%)		56 (11.7%)	20 (8.3%)	
Surgical Procedures			0.002			0.014 *
No	19 (3.4%)	0 (0.0%)		12 (2.5%)	0(0.0%)	
Yes	541 (96.6%)	280 (100.0%)		468 (97.5%)	240(100.0%)	
Radiotherapy/Surgery			<0.001			<0.001 **
No	332 (3.4%)	120 (42.9%)		249 (51.9%)	180 (75.0%)	
Yes	228 (96.6%)	160 (57.1%)		231 (48.1%)	60 (25.0%)	
Locoregional/Systemic Therapy			<0.001			0.867
No	402 (71.8%)	160 (57.1%)		317 (66.0%)	160 (66.7%)	
Yes	158 (28.2%)	120 (42.9%)		163 (34.0%)	80 (33.3%)	

** *p*-value < 0.001; * *p*-value < 0.05, calculated by simple chi-squared tests.

## Data Availability

Not applicable.

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
