# Peer review of "Risk Prediction of Second Primary Endometrial Cancer in Obese Women: A Hospital-Based Cancer Registry Study"

_ijerph, 2021, doi:10.3390/ijerph18178997_

Round 1
Reviewer 1 Report
The authors have written an interesting article. They appear to study second primary cancers associated with endometrial cancer using BMI stratification. The authors could improve the article as they address the following:
- More extensive literature reviews related to SPCs and tree-based methods
- Better connection between predictive modeling and CART-based subgroup analysis
- Better explanation about the CART-based subgroup analysis
- Distinguish between SPCs and SPEC
- Provide limitations and future directions of this study
Lines 34-41. The authors should elaborate more second cancers after EC including the relationship between EC and SPEC.
Line 54. In Table2, the authors should mention how they calculated p-values. Simple chi-squared tests?
Lines 64-88. There too many tree-based algorithms. The authors can consider KNN and Naïve Bayes too. There is no explanation about how the authors selected hyperparameters in each algorithm. For example, RF need to choose the number of trees, the number of predictors used, and the number of observations. Reproducibility is very important in data modeling. The authors should describe their procedure more detail.
Line 89. Why did CART produce an ideal prediction? Different measures produced different results. The authors need to elaborate their choice.
Lines 92-94. In Table 4 and Figure 1, it is not clear how the authors obtain the result. Are the results based on the entire data set or test data? To perform predictive modeling, the authors need a proper study design. May need better resolution for the figure.
Line 100. Figure 2 shows 11 terminal nodes but 13 subgroups are mentioned. Are they different? If so, please specify how 13 subgroups are obtained.
Line 130. Again, there is no explanation about how to calculate the p-values. Additionally, the authors can perform CMH test for 2x2x2 contingency table to compare between two BMI groups.
Line 172. Should 13 prognostic factors be 10? In Introduction, 10 factors are mentioned.
Author Response
Dear Reviewers,
Thank you for your valuable opinions. We so appreciated your patience.
We had reviewed and cited more articles related to SPCs and tree-based methods. We also corrected the errors in previous vision of the article. In this study, we only used 11 variables for obese women (BMI>25) for CART-based 11 subgroups analysis. In this study, we paired primary endometrial cancer (PEC) with second primary endometrial cancer (SPEC). Although there are different second primary cancers (SPCs) happened in primary endometrial cancer, we just evaluate the predictive factors for SPEC in this study only.
There might be some limitations from using only Taiwan’s local hospital registry data-base that could not represent other ethnicity. The tree-based algorithm was depended on local consensus to decide the variables in the predictive modeling. The international database pooling analysis was suggested for future study. Some clinico-pathological factors such as histological type, family history of cancer, timing of chemo-therapy exposure and regimen used should be included in the future analysis. The strength of our current study was a comprehensive Taiwan hospital registry database collecting. Our promising result could guide us to create another possible predictive modeling for other gynecologic cancers in the future.

Reviewer 2 Report
This study is the interesting and comprehensive summary for the understanding of the treated topic. The statistical tools used are sufficient for the purpose of the work and relevant to the achieved results. However, it is suggested that some points be reviewed.

Author Response
Dear Reviewers:
Thank you for your valuable opinions. We so appreciated your patience.
We had reviewed and cited more articles related to SPCs and tree-based methods. We also corrected the errors in previous vision of the article. In this study, we only used 11 variables for obese women (BMI>25) for CART-based 11 subgroups analysis. In this study, we paired primary endometrial cancer (PEC) with second primary endometrial cancer (SPEC) and 520 patients had SPEC. Although there are different second primary cancers (SPCs) happened in primary endometrial cancer, we just evaluate the predictive factors for SPEC in this study only. We made some mistakes in previous manuscript. The wrong finger1 was used. We had already replace with a new one and also improved the resolution.
There might be some limitations from using only Taiwan’s local hospital registry data-base that could not represent other ethnicity. The tree-based algorithm was depended on local consensus to decide the variables in the predictive modeling. The international database pooling analysis was suggested for future study. Some clinicopathological factors such as histological type, family history of cancer, and timing of chemotherapy exposure and regimen used should be included in the future analysis. The strength of our current study was a comprehensive Taiwan hospital registry database collecting. Our promising result could guide us to create another possible predictive modeling for other gynecologic cancers in the future.

This manuscript is a resubmission of an earlier submission. The following is a list of the peer review reports and author responses from that submission.